# Crystal structure of adenosine $A_{2A}$ receptor in complex with clinical candidate Etrumadenant reveals unprecedented antagonist interaction

Tobias Claff [1✉], Jonathan G. Schlegel [1], Jan H. Voss [1], Victoria J. Vaaßen [1], Renato H. Weiße [2], Robert K. Y. Cheng [3], Sandra Markovic-Mueller[3], Denis Bucher[3], Norbert Sträter [2] & Christa E. Müller [1✉]

The $G_s$ protein-coupled adenosine $A_{2A}$ receptor ($A_{2A}AR$) represents an emerging drug target for cancer immunotherapy. The clinical candidate Etrumadenant was developed as an $A_{2A}AR$ antagonist with ancillary blockade of the $A_{2B}AR$ subtype. It constitutes a unique chemotype featuring a poly-substituted 2-amino-4-phenyl-6-triazolylpyrimidine core structure. Herein, we report two crystal structures of the $A_{2A}AR$ in complex with Etrumadenant, obtained with differently thermostabilized $A_{2A}AR$ constructs. This led to the discovery of an unprecedented interaction, a hydrogen bond of T88[3.36] with the cyano group of Etrumadenant. T88[3.36] is mutated in most $A_{2A}AR$ constructs used for crystallization, which has prevented the discovery of its interactions. In-vitro characterization of Etrumadenant indicated low selectivity versus the $A_1AR$ subtype, which can be rationalized by the structural data. These results will facilitate the future design of AR antagonists with desired selectivity. Moreover, they highlight the advantages of the employed $A_{2A}AR$ crystallization construct that is devoid of ligand binding site mutations.

[1] PharmaCenter Bonn & Pharmaceutical Institute, Department of Pharmaceutical & Medicinal Chemistry, University of Bonn, An der Immenburg 4, 53113 Bonn, Germany. [2] Institute of Bioanalytical Chemistry, Center for Biotechnology and Biomedicine, University of Leipzig, Deutscher Platz 5, 04103 Leipzig, Germany. [3] leadXpro AG, PARK InnovAARE, 5234 Villigen, Switzerland. ✉email: tobias.claff@uni-bonn.de; christa.mueller@uni-bonn.de

G protein-coupled receptors (GPCRs) activated by the nucleoside adenosine are widely distributed and play important roles in transcellular signaling[1,2]. Four subtypes of adenosine receptors (ARs) exist, the preferentially $G_i$ protein-coupled $A_1$- and $A_3$ARs, and the $G_s$-coupled $A_{2A}$- and $A_{2B}$ARs[3]. Coupling to additional G proteins has been described, e.g., to $G_q$ proteins[4–6]. Adenosine acts as a "stop signal" resulting in strong anti-inflammatory and immunosuppressive effects, mediated by the $A_{2A}$- and $A_{2B}$AR subtypes[7]. Blockade of the $A_{2A}$AR is beneficial for several pathological conditions, in which adenosine-$A_{2A}$AR signaling is increased[8,9]. For example, the $A_{2A}$AR antagonist Istradefylline has been approved in Japan and the USA for the treatment of Parkinson's Disease[10]. Preclinical studies suggest major effects of $A_{2A}$AR antagonists against Alzheimer's Disease[11,12]. The $A_{2A}$AR, and later on also the related $A_{2B}$AR, both of which are expressed by immune cells and may be upregulated in cancer cells, have recently emerged as drug targets for the immunotherapy of cancer, constituting purinergic immune checkpoints[13].

Etrumadenant (3-[2-amino-6-[1-[[6-(2-hydroxypropan-2-yl) pyridin-2-yl]methyl]-4-yl]pyrimidin-4-yl]-2-methylbenzonitrile, also known as AB928) was developed as one of the first dual-acting adenosine $A_{2A}/A_{2B}$ receptor antagonists[14]. It constitutes a unique chemotype featuring a poly-substituted 2-amino-4-phenyl-6-triazolylpyrimidine core structure. The drug has entered clinical development and has been evaluated in several clinical phase I and phase II trials for the treatment of cancer[15]. Despite its advanced stage in drug development, the characterization of Etrumadenant is limited, and the exact drug–receptor binding mode is unknown.

Here, we determined the high-resolution crystal structure of Etrumadenant in complex with a thermostabilized $A_{2A}$AR construct comprising only two point mutations that do not interfere with ligand binding. The structure reveals unique binding pocket interactions of Etrumadenant including an interaction of its cyano group with T88[3.36]; to the best of our knowledge, this type of interaction has not been previously observed. For comparison, we also determined the high-resolution crystal structure of Etrumadenant in complex with a widely used $A_{2A}$AR construct that contains a T88[3.36]A mutation in the binding pocket ($A_{2A}$-StaR2-bRIL-A277S). The structural findings were complemented with an in-vitro pharmacological characterization of Etrumadenant at all AR subtypes. The compound was found to display high affinity in the low nanomolar range for $A_1$-, $A_{2A}$-, and $A_{2B}$ARs, and it potently blocked G protein activation by these subtypes. Structural data provided an explanation for the compound's lack of selectivity.

## Results and discussion

**Exploring the $A_{2A}$AR binding pocket of Etrumadenant using optimized crystallization constructs.** We previously developed an optimized $A_{2A}$AR crystallization construct designated $A_{2A}$-PSB1-bRIL, that contains a single point mutation (S91[3.39]K) inside the allosteric sodium binding pocket to stabilize the inactive conformation which significantly enhanced protein thermostability[16]. For the co-crystallization of Etrumadenant, we used the same modification but inserted an additional point mutation (N154[ECL2]A) to remove a putative glycosylation site on extracellular loop (ECL) 2 of the receptor. This construct is designated $A_{2A}$-PSB2-bRIL (PSB: Pharmaceutical Sciences Bonn, bRIL refers to thermostabilized apocytochrome $b_{562}$RIL[17]). Mutation of the asparagine in position 154 to either alanine or glutamine had previously been utilized to eliminate post-translational N-linked glycosylation of the $A_{2A}$AR, as protein glycosylation is expected to inhibit crystal growth due to microheterogeneity[18–20]. Evidence of N-linked glycosylation is missing, and N154[ECL2] is not surface-exposed in available $A_{2A}$AR crystal structures[21], indicating that the non-glycosylated form of the $A_{2A}$AR crystallizes predominantly. Here, we additionally employed sodium dodecyl sulfate polyacrylamide gel electrophoresis (SDS-PAGE) to demonstrate that $A_{2A}$-PSB1-bRIL (bearing the wild type (wt) N154[ECL2]) is still partially glycosylated, whereas $A_{2A}$-PSB2-bRIL (bearing an N154[ECL2]A mutation) had lost N-linked glycosylation (Fig. 1). Glycosylated proteins typically migrate more slowly in SDS-PAGE and generate higher molecular weight smearing[22]. Despite the fact

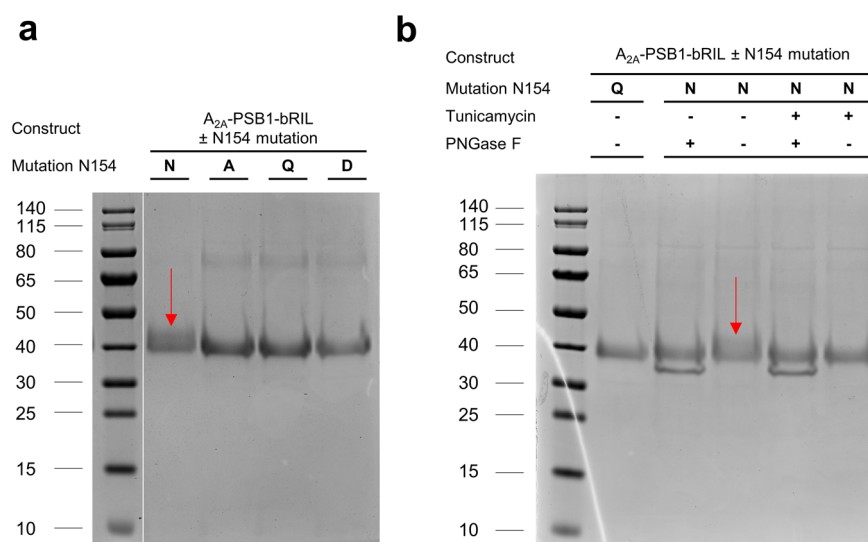

**Fig. 1 SDS-PAGE analysis of the $A_{2A}$AR glycosylation state. a** SDS-PAGE of $A_{2A}$-PSB1-bRIL compared to different N154 mutations in the same protein background to remove N-linked glycosylation. The red arrow points to the characteristic glycosylation smear. $A_{2A}$-PSB1-bRIL plus N154A corresponds to the crystallization construct $A_{2A}$-PSB2-bRIL. The protein marker originates from the same SDS-PAGE gel. **b** The effect of Tunicamycin and PNGase F on the SDS-PAGE mobility of $A_{2A}$-PSB1-bRIL, compared to N154Q in $A_{2A}$-PSB1-bRIL. Equal protein amounts were loaded onto the gel. The addition of Tunicamycin during $A_{2A}$-PSB1-bRIL expression or PNGase F treatment of the purified protein resulted in the removal of the characteristic glycosylation smear. The band for PNGase F ($\approx$36 kDa) is visible directly below the $A_{2A}$AR band (observed molecular weight $\approx$40 kDa, theoretical molecular weight $\approx$49 kDa). See Supplementary Fig. 1 for uncropped SDS-PAGE images.

**Table 1 Data collection and refinement statistics.**

| | A$_{2A}$-PSB2-bRIL-Etrumadenant (PDB ID 8C9W) | A$_{2A}$-StaR2-bRIL-A277S-Etrumadenant (PDB ID 8CIC) |
|---|---|---|
| *Data collection* | | |
| Space group | C222$_1$ | C222$_1$ |
| Cell dimensions | | |
| *a, b, c* (Å) | 39.16, 178.23, 139.60 | 39.36, 179.09, 140.57 |
| α, β, γ (°) | 90, 90, 90 | 90, 90, 90 |
| No. of reflections processed | 284,104 | 391,306 |
| No. of unique reflections | 21,413 | 29,632 |
| Resolution (Å) | 38.24–2.11 (2.32–2.1) | 46.86–2.10 (2.16–2.10) |
| Max. resolution aniso. (Å) | 2.16 (*a*$^*$), 2.11 (*b*$^*$), 2.50 (*c*$^*$) | not applied |
| $R_{merge}$ | 0.130 (1.697) | 0.172 (2.177) |
| $CC_{1/2}$ | 0.999 (0.529) | 0.999 (0.499) |
| $I / σI$ | 13.1 (1.3) | 11.1 (1.2) |
| Completeness spherical | 0.748 (0.159) | 1.000 (0.996) |
| Completeness ellipsoidal | 0.8980 (0.4125) | Not applicable |
| Redundancy | 13.3 (11.1) | 13.2 (13.4) |
| *Refinement* | | |
| Resolution (Å) | 38.24–2.11 (2.23–2.11) | 44.78–2.10 (2.14–2.10) |
| No. of reflections (test set) | 21,405 (994) | 55,857 (2840) |
| $R_{work}$ | 0.1965 (0.3298) | 0.1904 (0.3081) |
| $R_{free}$ | 0.2665 (0.5429) | 0.2144 (0.3764) |
| No. atoms (non-hydrogen) | | |
| A$_{2A}$AR | 2369 | 2349 |
| bRIL | 689 | 705 |
| Ligand | 32 | 32 |
| Lipids, polyethylene glycol (PEG) and waters | 284 | 604 |
| *B*-factors (Å²) | | |
| A$_{2A}$AR | 40.5 | 41.8 |
| bRIL | 70.8 | 76.7 |
| Ligand | 29.6 | 32.2 |
| Lipids, PEG and waters | 48.3 | 64.4 |
| Root-mean-square deviation (RMSD) | | |
| Bond lengths (Å) | 0.012 | 0.003 |
| Bond angles (°) | 1.305 | 0.55 |

$^*$For each structure, data from a single crystal was collected. The statistics for the highest resolution shell are shown in parentheses.

that only a single glycosylation site is present, smearing could be observed for A$_{2A}$-PSB1-bRIL, whereas a sharper band was detected for the N154$^{ECL2}$A mutant (A$_{2A}$-PSB2-bRIL) as well as for N154$^{ECL2}$Q and N154$^{ECL2}$D mutants, indicating the loss of glycosylation (Fig. 1). Alternatively, the glycosyl residues of A$_{2A}$-PSB1-bRIL could be cleaved off by the enzyme peptide-N-glycosidase F (PNGase F)[23] when added to the purified protein. Glycosylation could also be prevented during receptor expression by addition of the glycosylation inhibitor Tunicamycin[24].

Utilizing the optimized construct A$_{2A}$-PSB2-bRIL, we obtained the crystal structure of the A$_{2A}$AR in complex with Etrumadenant at 2.1 Å resolution (see Table 1 for detailed data collection and refinement statistics). Etrumadenant was well resolved within the orthosteric ligand binding pocket (Fig. 2a, b). Its scaffold shows unique interactions within the A$_{2A}$AR's orthosteric binding pocket. Importantly, the cyano group forms a direct hydrogen bond to T88$^{3.36}$ (N-O distance 2.8 Å) (Fig. 2a, b) representing an interaction that has so far not been observed in A$_{2A}$AR co-crystal structures with various antagonists. T88$^{3.36}$ is conserved within the AR family and was shown to be directly involved in A$_{2A}$AR agonist binding (illustrated for 5′-N-ethylcarboxamideadenosine (NECA) in Fig. 2c)[20]. It undergoes significant conformational changes during receptor activation[25]. The interaction of Etrumadenant with T88$^{3.36}$ by direct hydrogen bonding stabilizes the A$_{2A}$AR in its inactive state. Notably, the A$_{2A}$-StaR2-bRIL construct that is extensively used to determine inactive state A$_{2A}$AR crystal structures[16] harbors a T88$^{3.36}$A

mutation (see Fig. 3a), that can be expected to affect the affinity of Etrumadenant and possibly other antagonists. In fact, the affinity of Etrumadenant is ~47-fold lower for A$_{2A}$-StaR2-bRIL as compared to the wt A$_{2A}$AR (K$_i$ values of 39.8 nM compared to 0.85 nM, see Table 2). In contrast, the affinity of Etrumadenant for our optimized crystallization construct A$_{2A}$-PSB2-bRIL remained unaltered (K$_i$ 1.12 nM).

Besides the hydrogen bond to T88$^{3.36}$, Etrumadenant shows multiple additional receptor-ligand interactions. The phenyl ring of Etrumadenant is stabilized by π-π interactions to H250$^{6.52}$ (T-shaped) and W246$^{6.48}$ (stacked) (Fig. 2b). Its 2-methyl group comes in contact to V84$^{3.32}$, L85$^{3.33}$ and F168$^{ECL2}$. It is additionally exposed to a water network connecting the ligand to helices II and III (Fig. 2a). The 2-aminopyrimidine core is stabilized by π-π stacking interactions to F168$^{ECL2}$ (Fig. 2a) and forms key anchoring interactions by hydrogen bonding of the *N*3 and the exocyclic NH$_2$-group to N253$^{6.55}$ (Fig. 2a, b). Hydrogen bonding interactions of the side-chain of N253 are also observed for other ligands including agonists (Fig. 2c) and antagonists (see blue rectangles in Fig. 3). The hydrogen bond network is extended by a direct interaction of the exocyclic NH$_2$-group of Etrumadenant with E169$^{ECL2}$ (Fig. 2b). E169$^{ECL2}$ forms a salt bridge to H264$^{ECL3}$ that is frequently observed in A$_{2A}$AR crystal structures, but was found to be dependent on the structure of the antagonist and the pH value during crystallization[16].

The triazolyl ring of Etrumadenant (Fig. 2d), connected to the 6-position of the core aminopyrimidine, and bearing a substituted

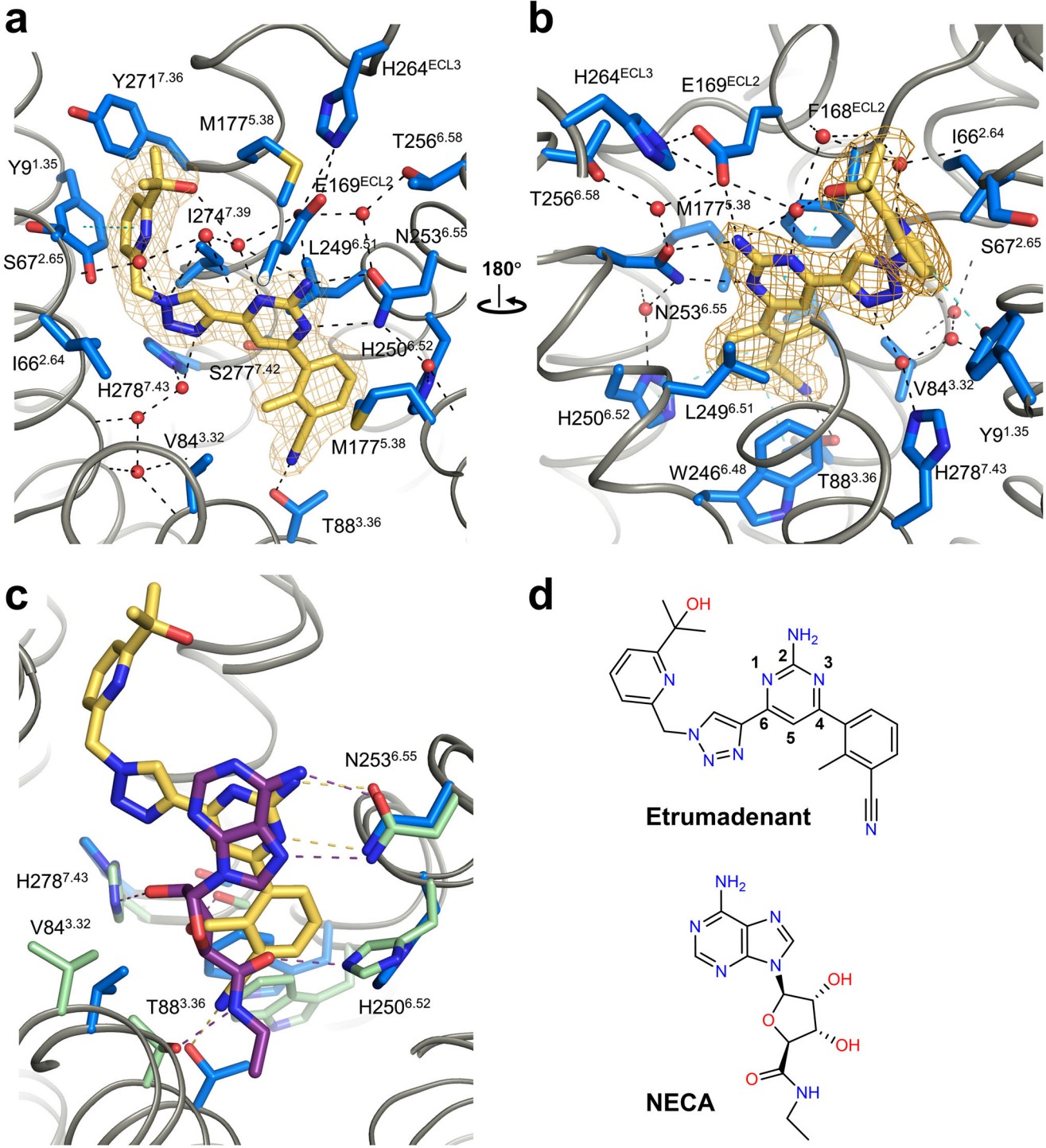

**Fig. 2 The A_{2A}AR binding pocket of Etrumadenant. a** Binding pocket of Etrumadenant with residues L167 and F168 clipped for enhanced visualization. The 2F_o–F_c electron density for Etrumadenant is shown in orange mesh (contoured at 1.0 σ). **b** Binding pocket of Etrumadenant rotated by 180° compared to (**a**) with parts of the ECL2 and residues A265 to M270 clipped. The 2F_o–F_c electron density for Etrumadenant is shown in orange mesh (contoured at 1.0 σ). Black dashed lines represent hydrogen bonds whereas cyan-colored dashed lines show π-π interactions. **c** Structural alignment of the A_{2A}-PSB2-bRIL-Etrumadenant (blue/yellow) binding pocket with that of NECA (PDB: 2YDV, represented in green/purple). Hydrogen bonds are shown in yellow and purple, respectively. **d** Chemical structures of Etrumadenant and NECA.

pyridylmethylene residue, forms π-π stacking interactions with F168^{ECL2} and water-mediated hydrogen bonding to H278^{7.43} and to the backbones of A59^{2.57}, I80^{3.28} and A81^{3.29} (Fig. 2a). The pyridine ring is located in close proximity to the entrance of the orthosteric ligand binding pocket at the extracellular ends of helices I and II with direct contacts to S67^{2.65} and Y271^{7.36}. The

2-hydroxyisopropyl residue that is attached to the pyridine of Etrumadenant shows three ambiguous rotamers. We chose to model the rotamer conformation with the hydroxy group in close proximity to a nearby water molecule thereby forming an intramolecular water-mediated hydrogen bond to the pyrimidine N1-nitrogen (Fig. 2a, b).

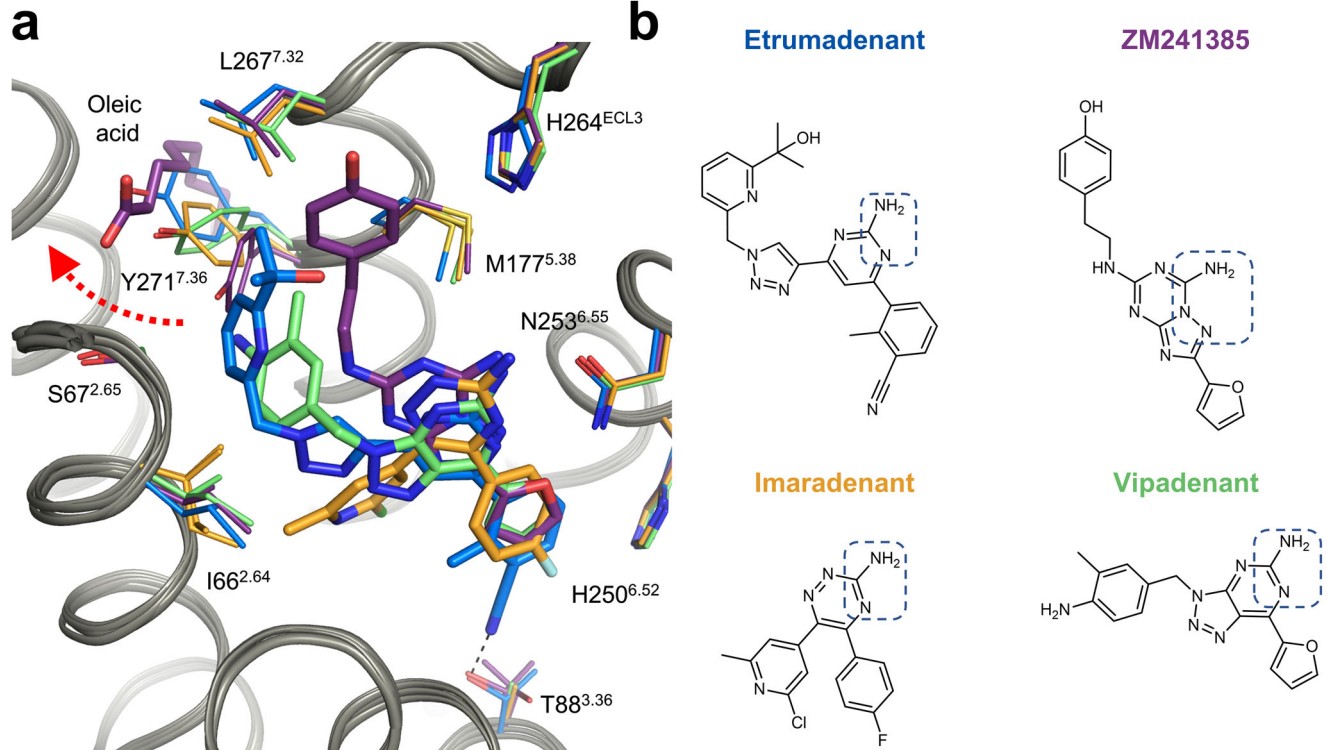

**Fig. 3 Comparison of the Etrumadenant binding pocket with that of selected A₂ₐAR antagonists. a** The binding pose of Etrumadenant (blue) is compared to the binding pockets of ZM241385 (purple, PDB ID 4EIY[21]), Vipadenant (green, PDB ID 5OLH[27]) and Imaradenant (orange, PDB ID 6GT3[26]). The red-colored dashed arrow represents the conformational movement of Y271[7.36] in the A₂ₐ-PSB2-bRIL-Etrumadenant structure. Of note: The structures of the Imaradenant- and the Vipadenant-complex have been obtained with the A₂ₐ-StaR2 construct that among other mutations contains a T88[3.36]A point mutation. The structure of the ZM241385-complex showed two different conformations for T88[3.36]. **b** Chemical structures of the depicted antagonists. The dotted blue rectangles highlight structural moieties that form the key hydrogen bonding anchor to N253[6.55].

**Table 2 Binding affinities of Etrumadenant and selected antagonists for the human adenosine receptors and for crystallization constructs[a].**

| Compounds | Human A₁AR | Human A₂ₐAR | Human A₂ᵦAR | Human A₃AR |
|---|---|---|---|---|
| | [³H]DPCPX (or [³H]CCPA) pK$_i$ ± SEM | [³H]MSX-2 pK$_i$ ± SEM | [³H]PSB-603 pK$_i$ ± SEM | [³H]PSB-11 (or [¹²⁵I]-AB-MECA[b]) pK$_i$ ± SEM |
| Etrumadenant | 8.12 ± 0.08 (8.15 ± 0.02) | 9.07 ± 0.14 | 8.50 ± 0.06 | 6.50 ± 0.14 |
| ZM241385 | 6.65[b] | 8.69 ± 0.20 | 7.53 ± 0.20 | (< 5.00[b]) |
| PSB-603 | (<5.00[c]) | <5.00[c] | 9.26[c] | <5.00[c] |
| Preladenant | (6.53[d]) | 9.05[d] | <6.00[d] | <6.00[d] |
| | **A₂ₐ-PSB2-bRIL** | **A₂ₐ-StaR2-bRIL** | | |
| | [³H]MSX-2 pK$_i$ ± SEM | [³H]MSX-2 pK$_i$ ± SEM | | |
| Etrumadenant | 8.95 ± 0.12 | 7.40 ± 0.05 | | |

[a]pK$_i$ values were determined as means from at least three independent experiments ± standard error of the mean (SEM) performed on CHO cell membranes expressing the respective human wt AR, or on *Sf9* insect cell membranes for the two crystallization constructs A₂ₐ-PSB2-bRIL and A₂ₐ-StaR2-bRIL. [³H]CCPA and [¹²⁵I]-AB-MECA represent agonist radioligands whereas all other radioligands are antagonists at ARs. [b]Ongini et al.[55]; [c]Borrmann et al.[29]; [d]Burbiel et al.[28].

The sidechain of Y271[7.36] was observed to be highly flexible when comparing different A₂ₐAR co-crystal structures[19,26]. It adapts the hydrophobic pocket to the size of the ligand (as depicted for a selection of ligands in Fig. 3). The relatively large Etrumadenant molecule requires a significant sidechain movement of Y271[7.36]. This sidechain is located much closer to the orthosteric binding pocket in the ZM241385-bound A₂ₐAR crystal structure, where it is hydrogen-bonded to the water network around the ligand (Fig. 3)[21]. In that structure, an additional oleic acid molecule occupies the space which Y271[7.36] adopts in the current Etrumadenant structure, where the hydrophilic head group of the oleate is displaced by the

rotation of Y271[7.36] (also compare structures of Imaradenant[26] and Vipadenant[27]) (Fig. 3).

Next, we additionally obtained the crystal structure of Etrumadenant in complex with a modified A₂ₐ-StaR2-bRIL receptor construct in which the S277[7.42]A mutation had been reverted to wt (designated A₂ₐ-StaR2-bRIL-A277S), but which still harbored the T88[3.36]A mutation in the binding pocket. A co-crystal structure could be obtained at the same high resolution of 2.1 Å (see Table 2 for detailed refinement statistics). Surprisingly, even though a major interaction partner of the ligand was mutated, the binding pockets of A₂ₐ-PSB2-bRIL-Etrumadenant

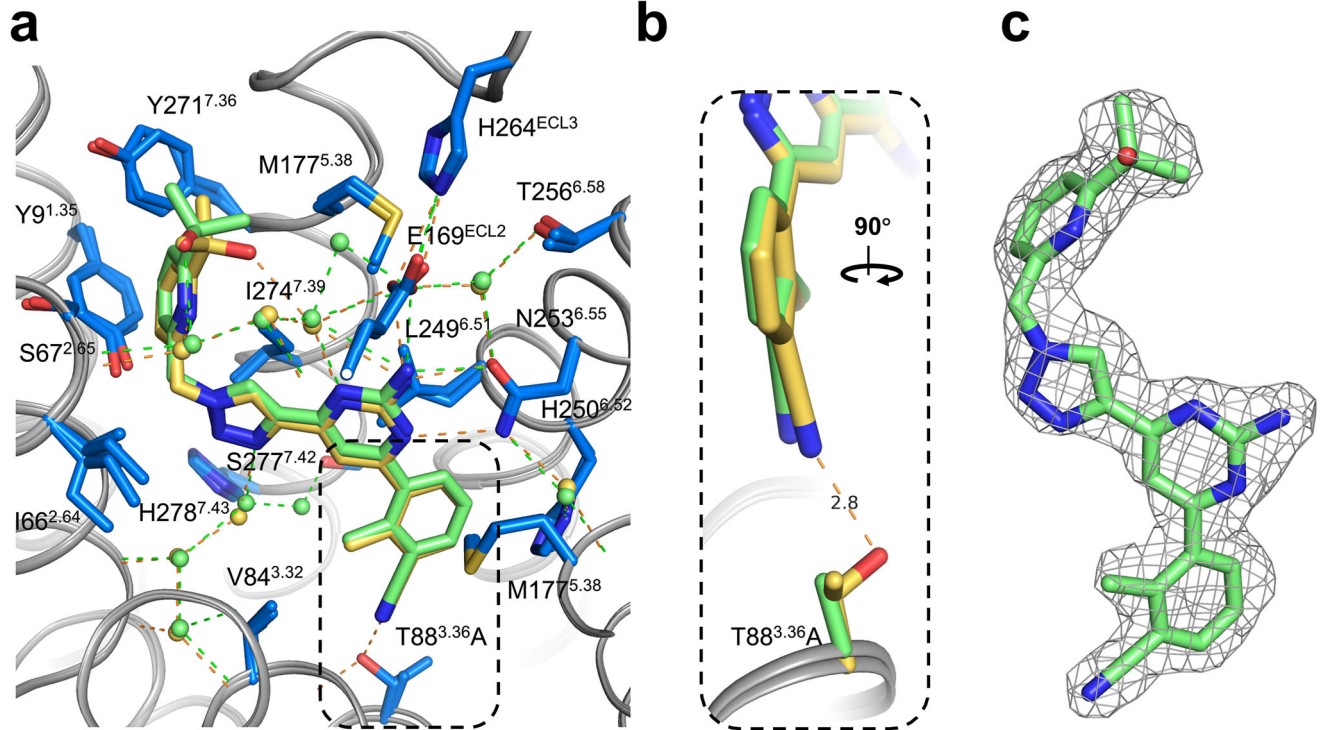

**Fig. 4 Comparison of the binding pockets of Etrumadenant in the A$_{2A}$-PSB2-bRIL and A$_{2A}$-StaR2-bRIL-A277S structures. a** The binding pose of A$_{2A}$-PSB2-bRIL-Etrumadenant (blue/yellow) is compared to the pose of A$_{2A}$-StaR2-bRIL-A277S-Etrumadenant (blue/green). Green- and yellow-colored dashes represent hydrogen bond interactions. **b** Zoomed panel highlighting the interaction of the cyano group with T88$^{3.36}$ at an N-O distance of 2.8 Å. **c** 2F$_o$–F$_c$ electron density for Etrumadenant in the A$_{2A}$-StaR2-bRIL-A277S structure, shown as gray mesh and contoured at 1 σ.

and A$_{2A}$-StaR2-bRIL-A277S-Etrumadenant are largely similar with only subtle differences (Fig. 4). Notably, the cyano group of Etrumadenant in the A$_{2A}$-PSB2-bRIL structure is slightly tilted, relative to the plane of the phenyl ring, towards the hydroxy group of T88$^{3.36}$ and deviates from the ideal planar orientation by ~8° (Fig. 4b). The same cyano moiety is planar in the T88$^{3.36}$A mutated structure, but is unable to form the same hydrogen bond interaction due to the mutation. Another difference between both structures can be identified in the rotamers of the 2-hydroxyisopropyl residue and the adjacent sidechain of Y271$^{7.36}$ (Fig. 4a) which confirms the initially observed flexibility of these moieties.

**Pharmacological characterization of Etrumadenant**. In the original patent describing Etrumadenant, affinity ranges were reported, but no specific K$_i$ or half-maximal inhibitory concentration (IC$_{50}$) values were provided[14]. In order to complement the pharmacological characterization of Etrumadenant, we determined its affinities for all human AR subtypes as well as for the crystallization constructs by radioligand binding assays (Table 2). To this end, we employed membrane preparations of Chinese hamster ovary (CHO) cells or *Spodoptera frugiperda* (*Sf9*) insect cells recombinantly expressing the respective AR subtype, or crystallization construct, respectively. Additionally, we investigated the inhibitory effects of Etrumadenant in G protein dissociation assays (Fig. 5).

In addition to its high affinity for the A$_{2A}$- and A$_{2B}$AR subtypes confirmed in the present study (K$_i$ values: A$_{2A}$, 0.851 nM; A$_{2B}$, 3.16 nM), we found that Etrumadenant also exhibits high affinity for the A$_1$AR (K$_i$ value: 7.59 nM versus the antagonist radioligand [$^3$H]DPCPX, and 7.08 nM versus the agonist radioligand [$^3$H]CCPA) (Table 2). Thus, the compound showed only about ninefold selectivity comparing A$_{2A}$- with A$_1$AR affinity, and only

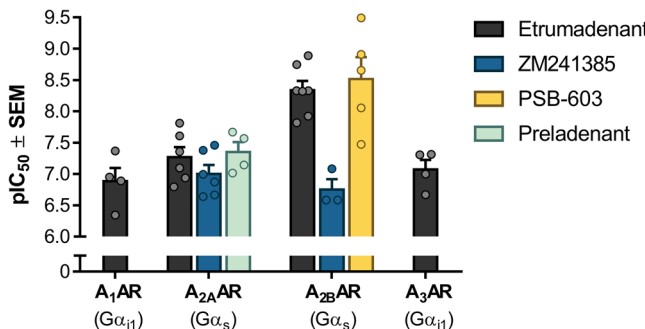

**Inhibition of G protein dissociation**

**Fig. 5 Inhibition of NECA-induced G protein dissociation by adenosine receptor antagonists.** pIC$_{50}$ values were determined as means from at least three independent experiments ± SEM using a BRET G protein dissociation assay[32]. Human embryonic kidney (HEK) cells were transfected with the respective AR and Gα-RLuc8, Gβ$_3$, and Gγ$_9$-GFP2 subunits. In the case of A$_1$- and A$_3$ARs, Gα$_{i1}$ was used whereas Gα$_s$ (short isoform GNAS-2) was used for the A$_{2A}$- and the A$_{2B}$ARs. The receptors were activated by NECA at its EC$_{80}$ for each receptor subtype (A$_1$AR 20 nM, A$_{2A}$AR 1 μM, A$_{2B}$AR 5 μM, A$_3$AR 20 nM), and concentration-dependent inhibition of the signal by Etrumadenant (or standard antagonists) was observed. EC$_{80}$ values depend on receptor expression[56] and probably also on G protein expression levels.

2-fold selectivity for the A$_{2B}$- versus A$_1$AR subtype. In contrast, we approved that Etrumadenant exhibits high selectivity versus the A$_3$AR (>100-fold), as determined in radioligand binding studies. For comparison, we determined the affinities of standard AR antagonists using the same assays (Table 2). While ZM241385 showed a moderate preference for the A$_{2A}$AR (K$_i$ values: A$_{2A}$, 2.04 nM; A$_{2B}$, 29.5 nM; 12-fold difference), the A$_{2A}$AR antagonist

Preladenant displayed similarly high $A_{2A}$ affinity as Etrumadenant ($K_i$: $A_{2A}$, 0.884 nM[28]) but showed high $A_{2A}$-selectivity. The $A_{2B}$AR antagonist PSB-603 was somewhat more potent than Etrumadenant ($K_i$: $A_{2B}$, 0.553 nM[29]) showing high selectivity for the $A_{2B}$AR subtype.

Subsequently, functional assays were performed to determine concentration-dependent antagonistic effects of Etrumadenant on receptor activation. To this end, we performed bioluminescence resonance energy transfer (BRET) based G protein dissociation assays employing *Renilla* Luciferase 8 (RLuc8) fused to Gα subunits and green fluorescent protein (GFP) attached to the Gγ subunit[30–32]. AR activation was induced with the non-selective agonist NECA at a concentration where it shows 80% of its maximal effect ($EC_{80}$). The preferentially $G_i$ protein-coupled $A_1$- and $A_3$AR subtypes were co-expressed with $Gα_{i1}$-RLuc8, $Gβ_3$, and $Gγ_9$-GFP proteins, whereas the Gs protein-coupled $A_{2A}$- and $A_{2B}$AR subtypes were co-expressed with $Gα_s$-RLuc8, $Gβ_3$, and $Gγ_9$-GFP proteins. Etrumadenant was able to block the activation of all four AR subtypes in a concentration-dependent manner. The antagonist was found to be most potent at the $A_{2A}$AR followed by the $A_{2B}$AR, but also showed significant antagonistic activity at the other AR subtypes, $A_1$ and $A_3$ (see Fig. 5). Blockade of the $G_i$ protein-coupled $A_1$- and $A_3$ARs will lead to an increase in intracellular cyclic adenosine monophosphate (cAMP) levels thereby counteracting the effects of antagonists at the $G_s$ protein-coupled $A_{2A}$- and $A_{2B}$ARs[33]. For this reason, $A_1$- and $A_3$ARs can be regarded as anti-targets in the development of AR antagonists for cancer therapy, and the lack of selectivity may contribute to side-effects[3].

For comparison, we additionally investigated the prototypical non-selective $A_{2A}/A_{2B}$AR antagonist ZM241385, the $A_{2A}$-selective antagonist Preladenant, and the $A_{2B}$-selective antagonist PSB-603. Preladenant inhibited the $A_{2A}$AR with similar potency as Etrumadenant in this assay ($IC_{50}$ values 85.1 nM, 53.7 nM), whereas the potency of ZM241385 ($IC_{50}$: 178 nM) was lower than that of Etrumadenant ($IC_{50}$: 4.57 nM) at the $A_{2B}$AR, but similar at the $A_{2A}$AR ($IC_{50}$ values: 100 nM; 53.7 nM). PSB-603 showed similarly high potency at the $A_{2B}$AR as Etrumadenant ($IC_{50}$ values: 3.02 nM; 4.57 nM). It should be kept in mind that the employed functional G protein activation assays require overexpression of receptors and G proteins[32]. Nevertheless, these data confirm that Etrumadenant is a potent antagonist of $A_{2A}$- and $A_{2B}$ARs, but its selectivity versus the $G_i$ protein-coupled ARs is low.

To explain this observation, we performed a sequence alignment of all AR subtypes and analyzed the conservation of amino acids that interact with Etrumadenant as observed in the $A_{2A}$AR co-crystal structures (Fig. 6). In fact, these amino acid residues are largely conserved in the $A_1$-, $A_{2A}$-, and $A_{2B}$AR subtypes, which is consistent with the high affinity of Etrumadenant for all three subtypes. The orthosteric binding pockets of the $A_{2A}$- and the $A_{2B}$AR differ only by one homologous amino acid exchange ($L249^{6.51}$ in the $A_{2A}$AR, $V250^{6.51}$ in the $A_{2B}$AR). The recently determined cryogenic electron microscopy structures of the $A_{2B}$AR in the active state confirmed a similar binding mode of the agonists adenosine and NECA in both receptor subtypes[34,35]. The extracellular ends of the $A_{2B}$AR are less conserved, and among the residues that are in contact with Etrumadenant in the $A_{2A}$AR two major differences can be observed: $L267^{7.32}$ and $Y271^{7.36}$ of the $A_{2A}$AR are exchanged for $K269^{7.32}$ and $N273^{7.36}$ present in the $A_{2B}$AR. Therefore, we hypothesize that Etrumadenant's aminopyridine core exhibits a comparable binding mode in the $A_{2B}$AR as in the $A_{2A}$AR, whereas the substituted pyridylmethylene residue, that extends towards the extracellular space and is relatively flexible, may show differences in binding at both $A_2$AR subtypes.

One notable difference between $A_1$- and $A_{2A}$ARs is $S67^{2.65}$ of the $A_{2A}$AR that is exchanged for $N70^{2.65}$ in the $A_1$AR. $S67^{2.65}$ forms

direct contacts with the pyridine core of Etrumadenant at the extracellular ends of the ligand binding pocket. The overall large conservation between the $A_1$- and $A_{2A}$AR ligand binding pockets substantiates our observation that Etrumadenant exhibits significant $A_1$AR affinity. However, the exchange of $S67^{2.65}$ to $N70^{2.65}$ in the $A_1$AR might affect the binding of Etrumadenant and explain the slightly reduced affinity of Etrumadenant for the $A_1$AR.

The $A_3$AR, on the other hand, shows significant differences being the least conserved AR subtype regarding Etrumadenant's binding pocket residues. Three hydrophobic amino acid residues that form direct Etrumadenant contacts in the $A_{2A}$AR ($I66^{2.64}$, $V84^{3.32}$, and $A273^{7.38}$) are exchanged in the $A_3$AR for different hydrophobic amino acids or glycine ($V72^{2.64}$, $L90^{3.32}$, and $G267^{7.38}$). Moreover, we observed direct interactions of Etrumadenant to the side chains of $E169^{ECL2}$ and $H250^{6.52}$ as well as a water-bridged hydrogen bond to $T256^{6.58}$. These residues are conserved among the $A_1$-, $A_{2A}$-, and $A_{2B}$ARs, whereas the $A_3$AR contains $V169^{ELC2}$, $S247^{6.52}$, and $I253^{6.58}$ in the analogous positions (Fig. 4, red boxes). Variation of these interacting residues in the $A_3$AR provides an explanation for the decreased affinity of Etrumadenant for the $A_3$AR.

## Conclusions

The adenosine receptor antagonist Etrumadenant represents a promising clinical candidate for the treatment of cancer, with high affinity for $A_{2A}$ and $A_{2B}$ARs. Both AR subtypes represent purinergic immune checkpoints inhibiting the immune system and showing ancillary direct cancer proliferation-enhancing and pro-angiogenic effects[7,33]. Blockade of $A_{2A}$- and $A_{2B}$ARs consequently is expected to exert anti-cancer activity. We show that the affinity of Etrumadenant for the $A_1$AR, which has been discussed as an anti-target in cancer therapy, is similarly high as for the $A_{2A}$- and $A_{2B}$AR, with a $K_i$ value in the single-digit nanomolar range. The first $A_{2A}$AR co-crystal structures of Etrumadenant in complex with the $A_{2A}$AR provide insights into the $A_{2A}$AR ligand binding pocket. They revealed that Etrumadenant stabilizes the inactive state of the $A_{2A}$AR by hydrogen bond interaction to $T88^{3.36}$ through its cyano group. A direct hydrogen bond to $T88^{3.36}$ has thus far not been observed in antagonist-bound $A_{2A}$AR crystal structures. Importantly, the $A_{2A}$-StaR2 construct that has been used for the vast majority of $A_{2A}$AR co-crystal structures[16] contains two mutations inside the ligand binding pocket ($T88^{3.36}$A and $S277^{7.42}$A) preventing the discovery of this hydrogen bond. In fact, out of the 24 different $A_{2A}$AR antagonists for which co-crystal structures have been solved to date (see Supplementary Table 1) only five have been determined with $A_{2A}$AR constructs harboring the native $T88^{3.36}$ (ZM241384[21], "cmpd-1"[36], PSB-2113[16], PSB-2115[16], and most recently KW-6356[37]). Nevertheless, the structure of a modified $A_{2A}$-StaR2-bRIL construct (that has the $S277^{7.42}$A mutation reverted to the wt residue) in complex with Etrumadenant revealed nearly identical binding poses of Etrumadenant, despite the $T88^{3.36}$A mutation. However, we show that the affinity of Etrumadenant for the $A_{2A}$-StaR2-bRIL crystallization construct is reduced by 47-fold compared to the wt $A_{2A}$AR ($K_i$ 39.8 nM vs. 0.851 nM) whereas the affinity to the employed optimized crystallization construct $A_{2A}$-PSB2-bRIL is unaltered ($K_i$ 1.12 nM). The discovered interaction with $T88^{3.36}$ will be relevant for the design and optimization of future $A_{2A}$AR antagonists as well as for dual $A_{2A}/A_{2B}$AR antagonists and pan-AR antagonists, in particular, since $T^{3.36}$ is conserved among the AR subtypes, not only in humans, but also in rats[38]. Its conservation likely also contributes to Etrumadenant's high affinity for the $A_1$AR. The $A_3$AR contains several non-conserved residues that are involved in $A_{2A}$AR binding, which could explain the selectivity of Etrumadenant versus the $A_3$AR subtype as observed in radioligand binding experiments

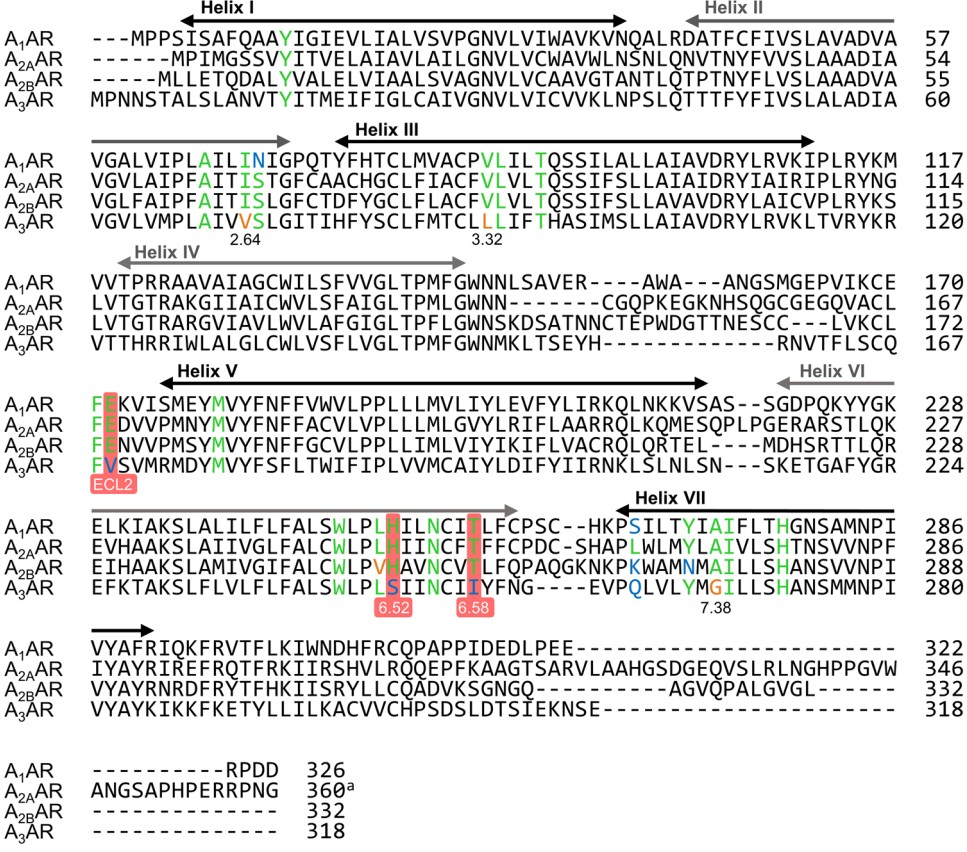

**Fig. 6 Adenosine receptor sequence alignment.** Colored residues indicate amino acids with direct contacts to Etrumadenant or interactions via one structural water molecule as observed in the $A_{2A}$-PSB2-bRIL-Etrumadenant structure. Green colored residues are conserved whereas blue colored residues highlight significant subtype differences. Orange residues indicate exchanges for amino acids with similar properties. Sequences were aligned with Clustal Omega. ᵃThe long C-terminal tail of the $A_{2A}$AR (residues 361–412) was omitted from the alignment.

(more than two orders of magnitude). The present X-ray structure will serve as a basis for the future design of tailored AR antagonists, which have great potential for the treatment of cancer as well as neurodegenerative diseases.

## Methods

**Expression, purification and crystallization of the $A_{2A}$-PSB2-bRIL-Etrumadenant complex.** The crystallization construct $A_{2A}$-PSB2-bRIL was cloned using site-directed mutagenesis in order to add the glycosylation mutation N154$^{ECL2}$A to the previously reported crystallization construct $A_{2A}$-PSB1-bRIL[16]. The N154$^{ECL2}$Q and N154$^{ECL2}$D mutations were cloned analogously. $A_{2A}$-PSB2-bRIL was expressed and purified in analogy to the procedure described for $A_{2A}$-PSB1-bRIL[16]. Briefly, $A_{2A}$-PSB2-bRIL was expressed in Sf9 insect cells using GP64-pFastBac1 as baculoviral expression vector. Cells were disrupted by osmotic shock and membranes were repeatedly washed using a washing buffer that contained a high amount of NaCl. The resuspended cell membranes were subsequently incubated with 50 µM Etrumadenant (obtained from MedChemExpress, cat. #HY-129393) and 2 mg mL$^{-1}$ iodoacetamide for 1 h prior to solubilization. $A_{2A}$-PSB2-bRIL was solubilized and purified from Sf9 membranes similarly as described for $A_{2A}$-PSB1-bRIL[16]. Etrumadenant was added to wash buffer I and to wash buffer II at 50 µM and 25 µM concentration, respectively. The protein was eluted with four column volumes of elution buffer containing 25 µM Etrumadenant, 25 mM HEPES pH 7.5, 800 mM NaCl, 10% (v/v) glycerol, 220 mM imidazole, 0.025% (w/v) dodecyl-β-D-maltoside (DDM), and 0.005% (w/v) cholesteryl hemisuccinate (CHS).

The $A_{2A}$-PSB2-bRIL-Etrumadenant complex was concentrated to a volume of 20–30 µL using 100 kDa cut-off Vivaspin concentrators (Sartorius), and immediately used for crystallization experiments. The complex was reconstituted into lipidic cubic phase using the two-syringe method[39] by mixing the protein with a molten lipid mixture [90% (w/w) 1-oleoyl-rac-glycerol (Sigma), 10% (w/w) cholesterol (Sigma)] in a 2 to 3 ratio. Crystallization experiments were performed using an automatic crystallization robot (Formulatrix NT8) by overlaying 50 nL of mesophase with 800 nL of precipitant solution. The $A_{2A}$-PSB2-bRIL-Etrumadenant complex crystallized in 30% (w/v) PEG400, 7% (w/v) Tacsimate pH 7.0 (Hampton Research, cat. #HR2-755)[40], 100 mM HEPES-Na pH 7.4, 1.8% (w/v) 2,5-hexandiol (Molecular Dimensions, cat. #MD2-100-226). Crystals were

harvested with micromounts (MiTeGen) and flash-frozen in liquid nitrogen without further cryo-protection.

**X-ray diffraction data collection and structure elucidation of the $A_{2A}$-PSB2-bRIL-Etrumadenant structure.** X-ray data collection was carried out at 100 K on EMBL beamline P14 of the DESY synchrotron (Hamburg, Germany). The x-ray wavelength used for the experiment was 0.97625 Å. An Eiger2 16 M detector was placed at a distance of 340 mm behind the crystal, which was rotated for 360° while diffraction images were recorded at 0.15° steps with exposure for 0.01 s. All datasets were indexed, integrated, scaled, and converted to structure factor amplitudes using ISPyB software: autoPROC[41], XDS[42], CCP4[43], POINTLESS[44], AIMLESS[45], STARANISO[46]. Crystallographic statistics are presented in Table 1. PDB ID 5IU4[47] was used as the starting model for refinement with phenix[48]. Coot[49] was used for model building. The stereochemical restraints for the ligand were generated with the GRADE web server[50]. The Ramachandran plot statistics were determined to 97.65% (favored), 2.09% (allowed), and 0.26% (disallowed).

**Expression, purification and crystallization of the $A_{2A}$-StaR2-bRIL-A277S-Etrumadenant complex.** $A_{2A}$ construct ($A_{2A}$-StaR2-bRIL-A277S) containing the same thermostabilizing and deglycosylation site mutations as PDB ID 5IU4[27] (except for the S277$^{7.42}$A mutation, which is reverted to wt) was codon-optimized and cloned between the BamHI and HindIII sites of pFastBac1 (Trenzyme). The bacmid was generated by transforming this plasmid into E. coli strain DH10EMBacY (MultiBac, Geneva Biotech). Isolated bacmid DNA was transfected into Sf9 insect cell using Cellfectin (Invitrogen) to generate baculovirus. For large-scale expression, High Five insect cells growing at 27 °C in Sf900 II medium (Invitrogen) at $2.5 \cdot 10^6 \cdot$ mL$^{-1}$ were infected with P2 baculovirus and harvested at 48 h post infection. Cells were harvested by centrifugation and the pellet was stored at −80 °C. Cells were thawed at room temperature and resuspended in 40 mM Tris-HCl pH 7.6, 1 mM ethylenediaminetetraacetic acid (EDTA), and cOmplete EDTA-free protease inhibitors tablets (Roche). Membranes were fractionated by passing the cell once using microfluidizer (Microfluidics) operated at 8000 psi. Membranes were centrifuged at 42000 rpm using a Ti45 rotor (Beckman) and washed once with 40 mM Tris-HCl pH 7.6, 1 M NaCl, and cOmplete EDTA-free protease inhibitors tablets. Membranes were centrifuged again and resuspended in 40 mM Tris−HCl

pH 7.6, cOmplete EDTA-free protease inhibitor cocktail tablets and frozen at −80 °C. Unless otherwise stated, all purification procedures were carried out at 4 °C. Membranes were solubilized with 1.5% (w/v) decylmaltoside (DM) and 0.1% (w/v) CHS for 1 h. Insoluble fractions were pelleted by centrifugation at 42,000 rpm using a Ti45 rotor (Beckman) for 30 min. $A_{2A}$-StaR2-bRIL-A277S was purified by loading the supernatant (supplemented with 8 mm imidazole) into a 5 mL cartridge containing Ni-NTA super flow resin (Qiagen) at 2 mL min$^{-1}$ using an ÄKTA pure system (Cytiva). The resin was first washed with 25 mL of 40 mm Tris-HCl pH 7.5, 400 mm NaCl, 0.15% (w/v) DM, 0.002% (w/v) CHS, and 8 mm imidazole and then washed once more with 25 mL of 40 mm Tris-HCl pH 7.5, 400 mm NaCl, 0.% (w/v) DM, 0.002% (w/v) CHS, and 72 mm imidazole. The protein was eluted in the same buffer containing 272 mm imidazole, concentrated using Vivaspin turbo 15 mL with a molecular weight cut-off of 50 kDa (Sartorius) and loaded onto a Superdex 200 10/300 GL increase column equilibrated in 40 mm Tris-HCl pH 7.4, 200 mm NaCl, 0.15% (w/v) DM, and 0.002% (w/v) CHS. Fractions containing $A_{2A}$-StaR2-bRIL-A277S were pooled, concentrated to 22.5 mg ml$^{-1}$ aliquots and frozen at −80 °C. Protein purity and homogeneity were controlled by SDS-PAGE and fluorescence size exclusion chromatography (FSEC).

For crystallization, frozen $A_{2A}$-StaR2-bRIL-A277S aliquots were thawed on ice and centrifuged at 18,213×g for 10 min. Lipidic cubic phase (LCP) crystallization was performed by mixing the protein into monoolein (containing 10% (w/w) cholesterol) at 2:3 (w/w) ratio. The resulting LCP was dispensed using the mosquito LCP (SPT Labtech) using a bolus/precipitant solution ratio of 40:800 nL. Crystals were obtained using precipitant solution containing 0.1 M sodium citrate pH 5.0, 50 mm sodium thiocyanate, 3% (v/v) 2-methyl-2,4-pentanediol (MPD), 21–32% (w/v) PEG400, and 2 mm theophylline. Crystals appeared overnight and grew to full size (50–60 μm in the longest dimension) over a week. To prepare the $A_{2A}$-StaR2-bRIL-A277S-Etrumadenant complex, crystals were soaked overnight in the same precipitant solution by replacing theophylline with 1 mm Etrumadenant[27]. After soaking overnight, crystals were harvested with MicroLoops LD (mitogen) and frozen in liquid nitrogen.

### X-ray diffraction data collection and structure elucidation of the $A_{2A}$-StaR2-bRIL-A277S-Etrumadenant structure.
Diffraction data were collected at the Swiss Light Source (SLS) beamline PXII. Crystal was exposed to a 25 × 13 μm X-ray beam (wavelength 0.99997 Å) at 25% transmission. A total of 180° of rotational data were collected using 0.1° oscillation and 0.05 s exposure per image. Dataset was processed and scaled to 2.1 Å using *XDS*[42] (built 20161205), then merged and converted to mtz file format with *AIMLESS*[45] (version 0.7.3 from *CCP4*[43] distribution 7.0.066). The structure was solved by molecular replacement with *PHASER*[51] (version 2.8.2 from *CCP4* distribution 7.0.066), using PDB ID 5IU4[27] as the search model. The model was rebuilt and refined using *COOT*[49] and *PHENIX*[52] (version 1.14) using TLS and optimizing 8CIC and ADP weight. After structure refinement, the model was validated using *MolProbity*[53] (from PHENIX version 1.14). The Ramachandran plot statistics were determined to 99.48% (favored), 0.52% (allowed), and 0% (disallowed).

### Radioligand binding studies.
Radioligand binding assays were performed on CHO cell membranes or *Sf9* insect cell membranes expressing the respective human wt adenosine receptor or $A_{2A}$AR crystallization constructs ($A_{2A}$-PSB2-bRIL or $A_{2A}$-StaR2-bRIL)[16,38]. The agonist [$^{3}$H]CCPA or the antagonist [$^{3}$H]DPCPX were employed as radioligands for the $A_1$AR (at 1 nm and 0.4 nm final concentration, respectively), the antagonist [$^{3}$H]MSX-2 was used for the $A_{2A}$AR (at 1 nm final concentration), the antagonist [$^{3}$H]PSB-603 for the $A_{2B}$AR (at 0.3 nm final concentration) and the antagonist [$^{3}$H]PSB-11 for the $A_3$AR (at 0.5 nm final concentration). All assays were performed in 50 mm Tris buffer (pH 7.4 at room temperature) at a final volume of 400 μL ($A_1$-, $A_{2A}$-, and $A_3$ARs) or 1000 μL ($A_{2B}$AR). Test compounds were dissolved in dimethyl sulfoxide and incubated at room temperature with the respective membranes and radioligand for 90 min ($A_1$AR, [$^{3}$H]CCPA), 60 min ($A_1$AR, [$^{3}$H]DPCPX), 30 min ($A_{2A}$AR), 75 min ($A_{2B}$AR) and 45 min ($A_3$AR). The final dimethyl sulfoxide concentration was 1%. Then, the mixture was filtered through GF/B glass fiber filters using a cell harvester (Brandel). For the $A_{2A}$AR assay, filters were pre-soaked in an aqueous solution of 0.3% (w/v) of polyethyleneimine for at least 30 min to reduce non-specific binding. Filters were then washed three times with 2 mL ice-cold Tris buffer (50 mm, pH 7.4 at room temperature). Filters containing the $A_{2B}$AR were washed with the same ice-cold Tris buffer but with the addition of 0.1% BSA. The remaining radioactivity was quantified after incubation for 9 h with scintillation cocktail (Beckmann Coulter) using a scintillation counter (Tricarb 2700TR).

### G protein dissociation assays.
TRUPATH BRET² assays[32] were performed as previously described[54] (TRUPATH was a gift from Bryan Roth (Addgene kit #1000000163)). In case of $G\alpha_{i/o}$-coupled adenosine receptors ($A_1$ and $A_3$ARs), a biosensor composed of $G\alpha_{i1}$-RLuc8, $G\beta_3$, and $G\gamma_9$-GFP2 was used. In case of the $G\alpha_s$-coupled $A_{2A}$- and $A_{2B}$ARs, the biosensor consisted of $G\alpha_s$-RLuc8, $G\beta_3$, and $G\gamma_9$-GFP2. Antagonist solution was incubated with the cells for 20 min before the addition of luciferase substrate solution (50 μm Deep Blue C, Biomol). Agonist solution (NECA) was added 5 min after the addition of the substrate solution at its EC$_{80}$ concentrations ($A_1$AR 20 nm, $A_{2A}$AR 1 μm, $A_{2B}$AR 5 μm, $A_3$AR 20 nm) and incubated for 5 min until measurement. GFP2 fluorescence (515 nm emission filter) was divided by RLuc8

luminescence (395 nm emission filter) to obtain BRET ratios. Data was normalized to controls (100% activation = NECA without antagonist, 0% activation = no agonist), and IC$_{50}$ values were obtained by a four-parameter sigmoidal curve fit in GraphPad PRISM v8.0 (GraphPad, San Diego, CA). The $G\alpha_s$ biosensor appeared to display a markedly lower expression level than the $G\alpha_i$ biosensor.

### SDS-PAGE analysis.
Proteins for SDS-PAGE were expressed and purified from 40 mL of *Sf9* insect cell culture for thermostability assessment[16]. Proteins were analyzed on homemade 10% SDS-PAGE gels casted using *bis*-2-amino-2-(hydroxymethyl)propane-1,3-diol (*bis*-Tris) buffer. Protein samples were prepared using NuPAGE loading dye (ThermoFisher, cat. #NP0007) supplemented with a final concentration of 50 mm dithiothreitol (DTT). Samples were incubated for 30 min at 37 °C prior to SDS-PAGE analysis using 3-(*N*-morpholino)propanesulfonic acid (MOPS) running buffer without addition of sodium hydrogen sulfite. SDS-PAGE gels were stained with Coomassie brilliant blue R-250 and destained using hot water. In order to investigate the effect of Tunicamycin on $A_{2A}$AR glycosylation, the respective insect cell culture was treated with 1 μg mL$^{-1}$ of Tunicamycin (CaymanChemical, cat. #11445) during infection. PNGase F (New England Biolabs, cat. #P0704S) was used to cleave the glycosylation in the purified protein prior to SDS-PAGE analysis using 375 units in a total reaction volume of 22.5 μL followed by incubation at 16 °C for 16 h.

## Data availability

The datasets generated during and/or analyzed during the current study are available from the corresponding author on reasonable request. The coordinates and structure factors for the obtained crystal structures have been deposited to the Protein Data Bank (https://www.rcsb.org/) under accession IDs 8C9W ($A_{2A}$-PSB2-bRIL-Etrumadenant) and 8CIC ($A_{2A}$-StaR2-bRIL-A277A).

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

## Acknowledgements

The authors thank Christin Vielmuth for her assistance with radioligand binding experiments. N.S. thanks the Deutsche Forschungsgemeinschaft for financial support (CRC 1423, project A6). The authors thank the team at Merck KGaA, Germany, for donation of AB928 X-ray structural data for A$_{2A}$-StaR2-bRIL-A277S-Etrumadenant used in the present work. This X-ray structure was originally generated by LeadXPro, contracted on behalf of Merck KGaA, Germany.

## Author contributions

T.C., J.G.S., J.H.V. and V.J.V.: investigation, methodology, validation, and formal analysis; R.K.Y., S.M.M. and D.B.: investigation, methodology, review and editing.; T.C.: conceptualization, writing—original draft, and visualization; R.H.W.: investigation and formal analysis, N.S.: investigation and supervision; C.E.M: project administration, conceptualization, supervision, funding acquisition, writing—review and editing.

## Funding

## Competing interests

The authors declare no competing interests.
