## [Peer Review File · Communications Chemistry]

Crystal structure of adenosine A2A receptor in complex with clinical candidate Etrumadenant reveals unprecedented antagonist interactionReviewers' comments:

Reviewer #1 (Remarks to the Author):

Claff et al. crystalized the A2A adenosine receptor in complex with Etrumadenant (AB928), a dual acting A2A/A2B antagonist currently in clinical trial. The important message conveyed in the manuscript is that many A2A structures are already available, but most are obtained with mutation of T88 (3.36). By comparing structures of T88 mutated and wild-type A2A receptors, the authors discovered an unprecedented interaction, a hydrogen bond of T88 with the cyano group of Etrumadenant.

This discovery is an important addition to the information from the currently available structures. It is known that T88 (3.36) is important for agonist binding, its interactions with A2A antagonists have not been well-documented. Since the T88 interaction with Etrumadenant is the major important discovery of this manuscript, it would be helpful to say how many of the currently available A2A structures with T88 mutated and how many not, although it is not practical to list every single one.

Also, since Etrumadenant was developed based on its dual interactions with A2A and A2B, and T3.36 is important agonist activation of both A2A and A2B receptors, the authors could explore or at least discuss the potential interactions of Etrumadenant with the A2B receptor based on its interactions with the A2A receptors and the recently available structures of the A2B receptor (Sci Adv. 2022 Dec 23;8(51):eadd3709. doi: 10.1126/sciadv.add3709; Cell Discov. 2022 Dec 28;8(1):140. doi: 10.1038/s41421-022-00503-1.)

Specific comments:

Line 65, "Structural data provided an explanation for the compound's lack of selectivity." Interactions of Etrumadenant with the A2B receptor could be explored or discussed.

Line 72-75, "previously developed an optimized A2AAR crystallization construct designated A2A-PSB1-bRIL, that contains a single point mutation (S913.39K) inside the allosteric sodium binding pocket to stabilize the inactive conformation which significantly enhanced protein thermostability" Does S91k mutation affect sodium effect on the A2A receptor?

Line 254-256: "The antagonist was found to be most potent at the A2BAR followed by the A2AAR... (see Figure 5)." But not as shown in Table 2.

Line 468: concentrations of radioligands not mentioned

Line 256-259: "Blockade of the Gi protein-coupled A1- and A3ARs will lead to an increase in intracellular cyclic adenosine monophosphate (cAMP) levels thereby counteracting the effects of the Gs protein-coupled A2A- and A2BARs."

Explain why cAMP increase counteracts the effects of the Gs protein-coupled A2A- and A2BARs?

Line 281-283: "and the A2BARs. The receptors were activated by NECA at its EC80 for each receptor subtype (A1AR 20 nM, A2AAR 1 μ M, A2BAR 5 μ M, A3AR 20 nM), and concentration-dependent inhibition of the signal by Etrumadenant (or standard antagonists) was observed." Why EC80 is high at A2A?

Reviewer #2 (Remarks to the Author):

The manuscript by Claff et al describes two crystal structures of the adenosine A2a receptor drug Etrumadenant in clinical development. In addition, the authors provide a comprehensive pharmacological characterisation that sheds light on the (underexplored) selectivity and functional activity profile of the drug.

The paper is concise and very well designed and executed, describing in detail the experiments performed and the results obtained. Of particular significance is the identification of a new interaction for the cyano group.

The manuscript provides new modelling data for the rational design of ligands for these receptors, alerting to the significant effect on the Etrumadenant A1 receptor.

In view of this referee the manuscript constitutes an excellent work and deserves to be published.

Reviewer #3 (Remarks to the Author):

The authors determined the high-resolution crystal structures of Etrumadenant in complex with two different A2A constructs, including a new thermostabilized A2AR construct that comprises only two point mutations. As a main finding, the structure reveals unique binding pocket interactions of Etrumadenant including a novel interaction of its cyano group 58 with T883.36. The T88 was mutated in alanine in A2A Star construct used in most studies reporting antagonist bound conformation of the A2A receptor but is not always engaged by antagonist. This article read well, the study is well conducted and describes a new interaction for stabilising the inactive state of the receptor bound to the antagonist Etrumadenant, although this interaction is not required for other antagonist. The structural analysis further explores the large binding pocket antagonist binding at A2A

1- There is a mistake in the legend of Figure 1, « The band for PNGase F (≈ 36 kDa) is visible directly below the A2AAR band (≈ 48 kDa). », the A2A has a molecular weight of 40 on the gel presented here (Line 106).

2- A2A-PSB2-bRIL, is certainly a new construct that combines only the mutation S91K and N154A, the last being part of a glycosylation site. However, this mutations has been previously used in several antagonist and agonist-bound A2A X-ray characterisation (Doré et al., 2011 ; Lebon et al 2011).

3- Line119, The authors should also cite original research article.

4- Line 165. Etrumadenant makes contacts with S67 and Y271. The interaction with S67 is interesting since CGS21680, an A2A agonist, can also establish a molecular contact with S67. Is the top part of TM2 in a more open conformation compared to other antagonist bound conformation?

5- Line 231 to 243. This paragraph is not clear, are the authors referring to previous reported affinity or affinity described here. Please clarify, what is the point being made here. This part is confusing.

Response to Reviewers' Comments

Reviewer #1 (Remarks to the Author):

Claff et al. crystalized the A_{2A} adenosine receptor in complex with Etrumadenant (AB928), a dual acting A_{2A}/A_{2B} antagonist currently in clinical trial. The important message conveyed in the manuscript is that many A_{2A} structures are already available, but most are obtained with mutation of T88 (3.36). By comparing structures of T88 mutated and wild-type A_{2A} receptors, the authors discovered an unprecedented interaction, a hydrogen bond of T88 with the cyano group of Etrumadenant.

This discovery is an important addition to the information from the currently available structures. It is known that T88 (3.36) is important for agonist binding, its interactions with A_{2A} antagonists have not been well-documented. Since the T88 interaction with Etrumadenant is the major important discovery of this manuscript, it would be helpful to say how many of the currently available A_{2A} structures with T88 mutated and how many not, although it is not practical to list every single one.

Thank you very much for your valuable suggestion. In fact, we have recently provided a list with all A_{2A}AR antagonist structures including their A_{2A}AR constructs (Claff et al. *Angew. Chemie Int. Ed.* 2022, DOI: <https://doi.org/10.1002/anie.202115545>, Table S1). Until then, the structure of only two A_{2A}AR antagonists was solved with a construct containing the native T88. Since then, the co-crystal structures of four more A_{2A}AR antagonists have been published (DOIs: 10.1021/acs.jmedchem.2c00462, 10.1016/j.ejmech.2022.114620, and 10.1124/molpharm.122.000633), and the only antagonist that was solved using a construct harboring the native T88 was with the antagonist KW-6356. In total, A_{2A}AR co-crystal structures with the native T88 so far only exist for A_{2A}AR antagonists ZM241385, "cmpd-1", KW-6356, and the two antagonist conjugates published by our group in 2022, PSB-2113 and PSB-2115. All other A_{2A}AR co-crystal structures with currently 19 different antagonists feature the T88A mutation present in the A_{2A}-Star2 construct.

Accordingly, we now provide an updated list of all A_{2A}AR crystal structures in complex with A_{2A}AR antagonists in Supplementary Table 1, and we added the following sentence to the manuscript:

"In fact, out of the 24 different A_{2A}AR antagonists for which co-crystal structures have been solved to date (see Supplementary Table 1) only five been determined with A_{2A}AR constructs harboring the native T88^{3,36} (ZM241384,²¹ "cmpd-1",³⁵ PSB-2113,¹⁶ PSB-2115,¹⁶ and most recently KW-6356³⁶)."

Also, since Etrumadenant was developed based on its dual interactions with A_{2A} and A_{2B}, and T3.36 is important agonist activation of both A_{2A} and A_{2B} receptors, the authors could explore or at least discuss the potential interactions of Etrumadenant

with the A2B receptor based on its interactions with the A2A receptors and the recently available structures of the A2B receptor (Sci Adv. 2022 Dec 23;8(51):eadd3709. doi: 10.1126/sciadv.add3709; Cell Discov. 2022 Dec 28;8(1):140. doi: 10.1038/s41421-022-00503-1.)

Thank you very much for this important suggestion. We have analyzed the amino acids that are in contact with Etrumadenant in the A_{2A}AR and compared these amino acids to the ones present in the A_{2B}AR. As a result, we suggest a similar binding mode for Etrumadenant to the A_{2B}AR but likely with subtle changes when it comes to the part of Etrumadenant that extends towards the extracellular space. We have included the analysis as well as the suggested references in the manuscript as follows:

“The orthosteric binding pockets of the A_{2A}- and the A_{2B}AR differ only by one homologous amino acid exchange (L249^{6,51} in the A_{2A}AR, V250^{6,51} in the A_{2B}AR). The recently determined cryogenic electron microscopy structures of the A_{2B}AR in the active state confirmed a similar binding mode of the agonists adenosine and NECA in both receptor subtypes.^{35,36} The extracellular ends of the A_{2B}AR are less conserved, and among the residues that are in contact with Etrumadenant in the A_{2A}AR two major differences can be observed: L267^{7,32} and Y271^{7,36} of the A_{2A}AR are exchanged for K269^{7,32} and N273^{7,36} present in the A_{2B}AR. Therefore, we hypothesize that Etrumadenant’s aminopyridine core exhibits a comparable binding mode in the A_{2B}AR as in the A_{2A}AR, whereas the substituted pyridylmethylene residue, that extends towards the extracellular space and is relatively flexible, may show differences in binding at both A₂AR subtypes.”

Specific comments:

Line 65, “Structural data provided an explanation for the compound’s lack of selectivity.”

Interactions of Etrumadenant with the A2B receptor could be explored or discussed.

We explored the potential binding mode of Etrumadenant to the A_{2B}AR in the section on selectivity (please also refer to the point discussed above).

Line 72-75, “previously developed an optimized A2AAR crystallization construct designated A2A-PSB1-bRIL, that contains a single point mutation (S913.39K) inside the allosteric sodium binding pocket to stabilize the inactive conformation which significantly enhanced protein thermostability”

Does S91k mutation affect sodium effect on the A2A receptor?

We would like to refer the reviewer to our referenced paper (Claff et al. Angew. Chemie Int. Ed. 2022, DOI: <https://doi.org/10.1002/anie.202115545>) where we compared the native sodium binding pocket with the S91K mutated one. The positively charged amine of the lysine sidechain perfectly mimics the sodium ion. Therefore, it can be

expected that the mutant is no longer responsive to sodium ions. We have not investigated the effect of sodium ions on A_{2A}AR ligand binding. However, while high concentrations of sodium inhibit the binding of A_{2A}AR agonists, no effect on the binding of A_{2A}AR antagonists are to be observed (Liu et al. Science 2012, DOI: 10.1126/science.1219218). The fact that the S91K-mutated construct does not bind to agonists anymore unfortunately prevents the investigation of the effects of sodium ions on agonist binding.

Line 254-256: “The antagonist was found to be most potent at the A_{2B}AR followed by the A_{2A}AR... (see Figure 5).” But not as shown in Table 2.

We thank the reviewer for careful reading and for detecting this error. We have corrected it accordingly:

“The antagonist was found to be most potent at the A_{2A}AR followed by the A_{2B}AR... (see Figure 5).”

Line 468: concentrations of radioligands not mentioned

The concentration of radioligands was added as follows:

“The agonist [³H]CCPA or the antagonist [³H]DPCPX were employed as radioligands for the A₁AR (at 1 nM and 0.4 nM final concentration, respectively), the antagonist [³H]MSX-2 was used for the A_{2A}AR (at 1 nM final concentration), the antagonist [³H]PSB-603 for the A_{2B}AR (at 0.3 nM final concentration), and the antagonist [³H]PSB-11 for the A₃AR (at 1 nM final concentration).”

Additionally, we have added a previously missing statement regarding the presoaking of the glass fiber filters in polyethyleneimine for the A_{2A}AR radioligand binding assay (to reduce non-specific binding):

“For the A_{2A}AR assay, filters were pre-soaked in an aqueous solution of 0.3 % (w/v) of polyethyleneimine for at least 30 min to reduce non-specific binding.”

Line 256-259: “Blockade of the G_i protein-coupled A₁- and A₃ARs will lead to an increase in intracellular cyclic adenosine monophosphate (cAMP) levels thereby counteracting the effects of the G_s protein-coupled A_{2A}- and A_{2B}ARs.” Explain why cAMP increase counteracts the effects of the G_s protein-coupled A_{2A}- and A_{2B}ARs?

We apologize for the lack of clarity. The blockade of A₁- and A₃ARs counteract the effects of antagonists (e.g. drug candidates) at the G_s-coupled A_{2A}- and A_{2B}ARs. We have rephrased the sentence:

“Blockade of the G_i protein-coupled A₁- and A₃ARs will lead to an increase in

intracellular cyclic adenosine monophosphate (cAMP) levels thereby counteracting the effects of antagonists at the G_s protein-coupled A_{2A}- and A_{2B}ARs.

Line 281-283: “and the A_{2B}ARs. The receptors were activated by NECA at its EC₈₀ for each receptor subtype (A₁AR 20 nM, A_{2A}AR 1 μM, A_{2B}AR 5 μM, A₃AR 20 nM), and concentration-dependent inhibition of the signal by Etrumadenant (or standard antagonists) was observed.” Why EC₈₀ is high at A_{2A}?

EC₅₀ values of agonists at GPCRs determined in G protein-dependent assays are dependent on the expression levels of receptors (Fujioka and Omori Drug Discov. Today 2012). A relatively low expression level of the A_{2A}AR and its G_{α_s} biosensor is likely responsible for the high EC₈₀ value of the agonist in the employed recombinant cell line. We added the following sentence:

“EC₈₀ values depend on receptor expression³⁵ and probably also on G protein expression levels.”

Additionally, we added a statement to the Methods section to explain differences in G-protein expression levels between the G_{α_s} and G_{α_i} biosensors:

“The G_{α_s} biosensor displayed a markedly lower expression level than the G_{α_i} biosensor.”

Reviewer #2 (Remarks to the Author):

The manuscript by Claff et al describes two crystal structures of the adenosine A_{2A} receptor drug Etrumadenant in clinical development. In addition, the authors provide a comprehensive pharmacological characterisation that sheds light on the (underexplored) selectivity and functional activity profile of the drug.

The paper is concise and very well designed and executed, describing in detail the experiments performed and the results obtained. Of particular significance is the identification of a new interaction for the cyano group.

The manuscript provides new modelling data for the rational design of ligands for these receptors, alerting to the significant effect on the Etrumadenant A₁ receptor.

In view of this referee the manuscript constitutes an excellent work and deserves to be published.

We thank the reviewer for careful reading and for appreciating our work.

Reviewer #3 (Remarks to the Author):

The authors determined the high-resolution crystal structures of Etrumadenant in complex with two different A_{2A} constructs, including a new thermostabilized A_{2A}R construct that comprises only two point mutations. As a main finding, the structure reveals unique binding pocket interactions of Etrumadenant including a novel interaction of its cyano group 58 with T883.36. The T88 was mutated in alanine in A_{2A} Star construct used in most studies reporting antagonist bound conformation of the A_{2A} receptor but is not always engaged by antagonist. This article read well, the study is well conducted and describes a new interaction for stabilising the inactive state of the receptor bound to the antagonist Etrumadenant, although this interaction is not required for other antagonist. The structural analysis further explores the large binding pocket antagonist binding at A_{2A}

1- There is a mistake in the legend of Figure 1, « The band for PNGase F (≈36 kDa) is visible directly below the A_{2A}AR band (≈48 kDa). », the A_{2A} has a molecular weight of 40 on the gel presented here (Line 106).

We thank the reviewer for noticing this lack of clarity. Here, 48 kDa refers to the theoretical molecular weight of the crystallized A_{2A}AR construct. However, membrane proteins and especially GPCRs typically run at lower molecular weights on SDS-PAGE gels, likely due to incomplete unfolding. We also noticed that ~49 kDa is more accurate for the theoretical molecular weight. Hence, we have clarified the legend as follows:

“The band for PNGase F (≈36 kDa) is visible directly below the A_{2A}AR band (observed molecular weight ≈40 kDa, theoretical molecular weight ≈49 kDa)”

2- A_{2A}-PSB2-bRIL, is certainly a new construct that combines only the mutation S91K and N154A, the last being part of a glycosylation site. However, this mutations has been previously used in several antagonist and agonist-bound A_{2A} X-ray characterisation (Doré et al., 2011 ; Lebon et al 2011).

We agree that is very important to acknowledge previous work; in fact, we already referenced it accordingly in lines 79-82 of the manuscript: “Mutation of the asparagine in position 154 to either alanine or glutamine had previously been utilized to eliminate post-translational N-linked glycosylation of the A_{2A}AR, as protein glycosylation is expected to inhibit crystal growth due to microheterogeneity.^{18,19}”

In addition to the reference of Doré et al. (19 in the manuscript), we have now also added the reference of Lebon et al. 2011.

3- Line119, The authors should also cite original research article.

We fully agree and thank the reviewer for this suggestion adding the reference of Lebon et al. (2011) accordingly.

4- Line 165. Etrumadenant makes contacts with S67 and Y271. The interaction with S67 is interesting since CGS21680, an A_{2A} agonist, can also establish a molecular contact with S67. Is the top part of TM2 in a more open conformation compared to other antagonist bound conformation?

We compared the conformation of TM2 observed in the structures of Etrumadenant, CGS21680, and the two prototypical A_{2A}AR antagonists ZM241384 (PDB 4EIY) and caffeine (PDB ID 5MZP). For the reviewer's convenience, we have generated a figure of the overlay shown below. We could not observe a conformation that is more open with regard to TM2 when compared to the ZM241384- or caffeine-bound structures. The existing Figure 3 has now been modified to display S67^{2.65} and to illustrate the similar conformations of the discussed antagonist-bound structures.

Revised Figure 3:

5- Line 231 to 243. This paragraph is not clear, are the authors referring to previous reported affinity or affinity described here. Please clarify, what is the point being made here. This part is confusing.

We apologize for not making this paragraph clear enough. Etrumadenant has not been properly characterized by radioligand binding studies in any publication before. We have rephrased the sentences to emphasize that this is newly produced data:

“In addition to its high affinity for the A_{2A}- and A_{2B}AR subtypes confirmed in the present study (K_i values: A_{2A}, 0.851 nM; A_{2B}, 3.16 nM), we found that Etrumadenant also exhibits high affinity for the A₁AR (K_i value: 7.59 nM versus the antagonist radioligand [³H]DPCPX, and 7.08 nM versus the agonist radioligand [³H]CCPA) (Table 2). Thus, the compound showed only about 9-fold selectivity comparing A_{2A}- with A₁AR affinity, and only 2-fold selectivity for the A_{2B}- versus A₁AR subtype. In contrast, we approved that Etrumadenant exhibits high selectivity versus the A₃AR (>100-fold), as determined in radioligand binding studies.”

Additional changes

1. As requested by the journal, we modified Figure 5 showing single data points in addition to the mean values \pm SEM.
2. We optimized the drawing of NECA in Figure 2d to better align it with the structure in Fig. 2c.

Revised Figure 2:

REVIEWERS' COMMENTS:

Reviewer #1 (Remarks to the Author):

The revised version of the manuscript is acceptable for publication.

Reviewer #3 (Remarks to the Author):

The authors answered all my comments. Very nice work.